

# Ethylene Oxide Monitor with Part-per-Trillion Precision for In-Situ Measurements

Tara I. Yacovitch, Christoph Dyroff, Joseph R. Roscioli, Conner Daube, J. Barry McManus, Scott C. Herndon

Aerodyne Research, Inc., 45 Manning Road, Billerica, Massachusetts, 01821 USA

*Correspondence to*: Tara I. Yacovitch (tyacovitch@aerodyne.com)

**Abstract.** An Aerodyne Tunable Infrared Direct Absorption Spectrometer with 413 meter cell for the detection of ethylene oxide (EtO) is presented (TILDAS-FD-EtO). This monitor achieves precisions of <75 ppt or <0.075 ppb in 1 second and < 20 ppt in 100 seconds (1-sigma). We demonstrate precisions averaging down to 4 ppt in an hour (1-sigma precision) when

operated with frequent humidity-matched zeroes. A months-long record of 2022 ambient concentrations at a site in the Eastern United States is presented. Average ambient EtO concentration is on the order of 18 ppt (22 ppt standard deviation). Enhancement events of EtO lasting a few hours are observed, with peaks as high as 600 ppt. Back trajectory simulations suggest an EtO source nearly 35 km away. This source along with another are confirmed as emitters through mobile near-source measurements, with downwind concentrations in the 0.5 ppb to 700 ppb range depending on source identity and distance

downwind.

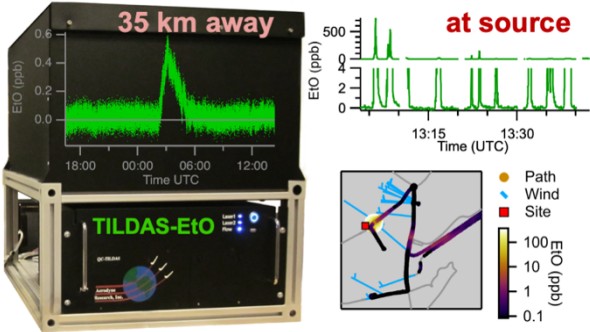

## 1 Introduction

Ethylene oxide (EtO, also known as EO or oxirane) is a reactive hydrocarbon commonly used in chemical manufacturing of polymers and glycols. It is also used to sterilize medical equipment (e.g. pacemakers, surgical kits) that cannot be exposed to

heat or humidity. Due to its reactivity, ethylene oxide is a carcinogen. The United States Environmental Protection Agency (EPA), through its Integrated Risk Information System (IRIS), has set an inhalation unit risk for EtO at $3.0 \times 10^{-3}$ per $\mu g/m^3$ ($5.5 \times 10^{-3}$ per ppb), for adult increased cancer risk based on human data (US EPA, 2016). Workplace limits for 8-hour and acute 5-min exposures are several orders of magnitude higher, on the order of 1 – 5 ppm (OSHA, 2002). The toxicity of this



chemical makes accurate, high precision measurements of ambient and near-source concentrations imperative; this advance is described herein.

Background levels of EtO are challenging to measure via extractive methods such as canister sampling. EtO can be formed during storage in the canisters used (Hoisington and Herrington, 2021; US EPA, 2019; Hasegawa, 2001). The levels of reported EtO formation are on the order of hundreds of ppt. Hoisington and Herrington (2021) note EtO formation in blanks filled with humidified air but not dry air or inert gas, and thus hypothesize the reaction to be between larger hydrocarbons and oxygen, catalyzed by the presence of water and metal surfaces. Both canister type/coating (US EPA, 2019) and canister cleanliness/cleaning protocol (Hoisington and Herrington, 2021) are thought to impact EtO formation.

Reported background concentrations of EtO at select US National Air Toxics Trends monitoring sites (NATTS) for the Oct 2018 – March 2019 period average 0.297 μg m$^{-3}$ and range between 0.185 – 0.397 μg m$^{-3}$ (103 to 220 ppt) (US EPA, 2019). More recent EPA data from 2022 at Massachusetts measurement sites show 24-hour concentrations between 0 and 0.270 μg m$^{-3}$ (0 – 150 ppt) (US EPA, 2022a) (data accessed 2022/8/30). Olaguer et al. (2020) report near-source 24 hr average concentrations in the 0.42 – 76.0 μg m$^{-3}$ range (233 ppt – 42.2 ppb), the lower value representative of ambient background, and the higher value sampled near a vent at a sterilization facility. These measurements all rely on canister sampling methods. To date, this study is the first in-situ measurement of real EtO sources in the published literature.

Several additional in-situ instruments for the detection of EtO have been developed recently. Gupta et al. (2022) describe a cavity-enhanced absorption spectrometry method with precision < 1 ppb (1 sigma, 60 seconds) and 0.5 ppb (1 sigma, 15 minutes). Picarro, Inc. (2021) has publicized cavity ringdown spectroscopy (CRDS) instruments with detection limits of 0.1 – 0.25 ppb (3 sigma, 300 seconds) depending on instrument model. Entanglement Technologies (2022) lists a CRDS instrument with EtO detection at the ppb-level in 5 seconds with other VOCs detected, and at the ppt-level in 15 minutes in "lab-scan" mode. Aeris Technologies (2022) describes a laser-based EtO with 0.5 ppb sensitivity (1 sigma, 1 second). Here, we describe a commercially available Aerodyne EtO monitor (Aerodyne Research Inc., 2022b) based on direct-absorption spectroscopy that is capable of < 0.075 ppb precision at 1-second (1 sigma) and 0.020 ppb precision at 100 seconds (1 sigma). With frequent zeroing and data averaging we demonstrate a precision of < 4 ppt (1 sigma, 1 hr). Instrument performance and calibration is described. A months-long ambient EtO record at a site in Billerica, Massachusetts, USA is described, and enhancements are traced back to a potential inventory EtO source. This source, and another are confirmed via near-field mobile measurements.

## 2 Experimental

### 2.2 Instrument Description

The basis of our EtO monitor is our commercially available dual-laser tunable infrared direct absorption spectrometer (TILDAS-FD) platform (Aerodyne Research Inc., 2022a), which in this case is equipped with a single mid-infrared interband-cascade laser (nanoplus GmbH). For the system described herein, we use a multipass cell with 413 m optical pathlength and



an active volume of 1.8 liters for continuous flow applications. Details of the optical setup and flow system are described in
the SI.

We measure EtO in a narrow wavelength window near 3065 cm⁻¹ (3.26 μm), Figure 1. In total, more than 250 individual
absorption lines across 6 molecular absorbers are included in the spectroscopic fit: EtO (114 lines), water ($H_2O$, 18 lines),
formaldehyde (HCHO, 23 lines), ethane ($C_2H_6$, 28 lines), methane ($CH_4$, 12 lines), ethylene ($C_2H_4$, 56 lines). Methanol can be

included in the fit optionally (32 lines). Centre wavelengths, linestrengths, and broadening coefficients of all molecules except
EtO and methanol are from the HITRAN database (Gordon et al., 2017). The high-resolution line parameters for EtO at 3065
cm⁻¹ were derived at Aerodyne. Initial knowledge of absorption at this wavelength was gained from high-resolution Fourier
transform spectra by Lafferty et al. (2013). A measured spectrum at high-concentrations of EtO is shown in the SI, Figure S1.
Methanol lines are based on experiments by Harrison et al. (2012).

**2.3 Calibration and Zeroing**

The EtO measurement is based on a set of experimentally acquired absorption lines. These experiments were done on a
prototype TILDAS instrument with 76 m pathlength absorption cell, operating at 30 Torr. The absorption linestrengths were
calibrated in February 2020 using a certified EtO standard (Apel Riemer, certified value 0.1023 ppm, August 2019) determined
by P. Kariher at the US EPA to show good relative agreement (within 7%) among 18 tanks from 5 vendors (Kariher, 2022).

Pressure-dependent EtO line broadening and other changes in instrument setup such as the inlet may lead to additional
uncertainty or bias when operating the 413 m instrument at 20 Torr, and so additional calibrations are done regularly for this
instrument.

Calibrations are performed by quantitative dilution of high-concentration EtO standards, to achieve a multi-point calibration
curve (see SI, Figure S2). We find dry calibrations prone to long time constants, which we tentatively attribute to surface

effects. Humid standard additions are preferred, as they most closely resemble sampling conditions.

We use a 2021 Airgas calibration standard, containing EtO (1.075 ppm ± 5%) and ethane (1.092 ppm ± 5%) in a balance of
nitrogen (see SI, Figure S3). The inclusion of ethane in the calibration tank provides a secondary known species measurable
by the instrument and not prone to reactivity or inlet effects. The average calibration factor for a set of standard addition
calibrations performed over a representative week-long period is $m = 0.981 \pm 0.045$ (95% error bars). This calibration factor

implies 1 ppb of measured EtO would be corrected to 1.02 ppb EtO. However, we do not apply this small 2% correction to the
data, given a certified tank uncertainty of 5% and the 4.6% error bars on the average calibration factor.

Uncertainties in the certified values of commercially available calibration tanks is of concern for accurate calibration of this
and other EtO methods. A total of 4 commercially available standards have been measured by the TILDAS-FD-EtO monitor
described here, varying in vendors, and at nominal concentrations of 1 ppm except where noted. Their retrieved concentrations

deviated from their certified values by -2% (the above EtO and ethane standard), +9%, -417% (standard at 0.5 ppm) and +18%.
Spectral backgrounding (or autobackgrounding) is done by intermittently and regularly measuring air free of EtO. The acquired
background spectrum is used to divide subsequent sample spectra, reducing the impact of drift due to instrumental effects like



optical fringes and spectral baseline effects. The use of scrubbed air provides a near-humidity match between sample and background spectra, effectively flattening out the curvature of the baseline present under the EtO lines due to strong

neighboring water absorptions. Laboratory experiments suggest scrubber breakthrough on the scale of 3% is possible (3-5 SLPM flow rates) at high mixing ratios (hundreds of ppb). Indeed, mobile near-source measurements have shown such breakthrough when an autobackground occurs within a high-concentration plume. Correction of this data is possible after-the-fact by manually offsetting baselines or performing a spectral refit of the data.

The frequency of autobackgrounds is chosen to match the sampling strategy: mobile measurements aimed at capturing plumes

(brief enhancements over background) use a 5- to 15-minute autobackground cycle to reduce interference with plume transects; stationary sampling uses a 2- to 5-minute cycle to optimize long-term averaging.

## 3 Results

### 3.1 Instrument Performance

Precision for the TILDAS-FD EtO monitor at 1-second are < 70 ppt (1-sigma), regardless of stationary or mobile

measurements.

Figure 2 compares stationary and mobile ambient measurement Allan-Werle variance plots (Werle, 2011). Blue traces show stationary performance, with best precisions achieved when stationary by altering humidity-matched zeroes with ambient measurements every 2 minutes for a 50% duty cycle. Measurements average down well, from a base precision of 44 ppt (1-sigma at 2-sec) reaching 13 ppt at 2-minutes, 6.0 ppt at 15 minutes and 4.1 ppt at 1 hour (all precisions at 1-sigma).

The TILDAS-EtO monitor has also been used for near-source mobile monitoring, with less frequent autobackgrounds (5- to 10-minute frequencies). The instrument shows sensitivity to truck motion, particularly quick turns or stops which manifest as negative deviations in mixing ratio on the order of 0.5 ppb. Optical alignment minimizes but does not eliminate these effects, which are largely attributed to strain on the laser focusing objective. Continuous vibrations are less impactful. Performance while in motion on the highway is shown in Figure 2 (red traces). For these measurements, the instrument was mounted in the

Aerodyne Mobile Laboratory in a vibration-isolated rack and operated with a 10-minute humidity-matched zeroing cycle. The 1-second precision of 50 ppt averages to 28 ppt in 2 minutes.

### 3.2 Ambient Measurements

A months-long record of ambient EtO in Billerica, Massachusetts, USA was acquired (Figure 3), spanning winter, spring, and summer 2022. Averaging the hourly data for the entire period (with standard deviation in parentheses) yields Avg (SD) = 18

(22) ppt. Hourly averages for summertime data are less noisy than wintertime data due to the more aggressive zeroing cycle. Summertime concentrations of 33 (13) ppt appear slightly elevated compared to winter and spring averages (9 Feb – 30 April) 12 (23) ppt measurements. The averages are different at the 95% confidence level using Gaussian statistics and standard error of the mean (see SI, Table S3). These data are consistent with recent data reported by the EPA for 4 Massachusetts sites (US





EPA, 2022a): 2022 observations accessed 8/20/2022 range between 0 and 0.270 μg m$^{-3}$ (0 – 150 ppt) with a median of 0.090 μg m$^{-3}$ (50 ppt); they are below 2019 levels shown for EPA NAATS sites in New York and Pennsylvania (US EPA, 2019) of 0.298 – 0.361 μg m$^{-3}$ (165 – 201 ppt), though the EPA has since noted that true background concentrations are unknown due to the influence of canister artifacts (US EPA, 2021).

Several distinct EtO enhancement events are evident in the ambient record. One such event on 3/27/2022 is shown in Figure 4. This figure shows two plumes, the larger of the two reaching concentrations of 500 ppt, and lasting 3-4 hours near midnight

local time. No EtO activity (e.g., calibrations) was occurring in the lab during this week. During these winter and spring roof-top measurements, the EtO monitor briefly switches to laboratory air prior to humidity-matched autobackgrounds, providing several seconds of indoor air sampling. The laboratory air shows an "echo" of the outdoor EtO event ~3 hours delayed, which we attribute to the building's ventilation system gradually mixing in outdoor air.

Back-trajectory simulations for this event were performed using NOAA's HYSPLIT engine (Rolph et al., 2017; Stein et al.,

2015) (see Figures S5 – S6). These simulations suggest that regional transport was from the south-west during this time. This trajectory passes over a commercial sterilization facility approximately 35 km away that is known by the EPA to use EtO (US EPA, 2022b). In the following section, we describe near-field mobile measurements of this source showing clear EtO enhancements downwind. These ambient measurements highlight the benefits of the high-precision TILDAS-FD-EtO sensor over alternative methods like canister sampling, which typically have long integration times (24 hours) that would wash out

brief events and are prone to sampling artifacts at 100's of ppt levels (US EPA, 2021; Hoisington and Herrington, 2021).

## 3.3 Near-Field Mobile Measurements

Motivated by the sporadic enhancement events in the ambient measurement record, mobile measurements of two commercial sterilization facilities in Massachusetts (US EPA, 2022b) were conducted in August 2022. The first source visited, "Facility A", was the facility identified through Hysplit trajectory explorations of the 3/27/2022 event. Facility A was visited over the

course of ~4 hours, split between morning and afternoon. Average downwind concentrations are summarized in Figure 5, showing clear enhancements above background downwind of the facility. Concentration enhancements ~600 m from the source were around 5 ppb, with enhancements as high as 300 ppb measured 35 m from the facility. Additional transects, time series and spatial averages are shown in the SI.

The second source measured, "Facility B", is also a commercial sterilization facility (US EPA, 2022b), and is located 15 km

miles south of the Billerica MA stationary measurement site. The EPA has conducted a risk assessment of this facility and found enhanced cancer risk (US EPA, 2022b). Facility B also showed enhancements above background on this measurement day (maximum of 7.5 ppb 60 m downwind), though at far lesser concentrations than Facility A. Further details are presented in the SI.



## 4 Conclusions

The TILDAS-FD-EtO monitor achieves precisions of <75 ppt or <0.075 ppb in 1 second and < 20 ppt in 100 seconds (1-sigma precisions), with averaging down to 4 ppt in an hour (1-sigma) when operated with frequent humidity-matched zeroes. Ambient measurements at a Massachusetts site reveal EtO concentrations on the order of 18 ppt (22 ppt standard deviation). Distinct EtO events lasting a few hours are observed in the ambient record, with back trajectory simulations suggesting an EtO source nearly 35 km away. Mobile measurements directly downwind of this medical sterilization facility, as well as another

sterilization facility in the state, confirm the presence of EtO emissions at both sites, with downwind concentrations in the 0.5 ppb to 700 ppb range depending on source identity and distance downwind. These measurements highlight how continuous in-situ EtO monitoring with a high-precision sensor can provide information leading directly to EtO point source identification.

### Data availability

Ambient ethylene oxide dry air mixing ratios, hourly averages and mobile measurement data is publicly and freely available
at https://osf.io/jeywd/?view_only=421c0d48e2d1449488c14f883a7859b6

### Author Contribution

The manuscript was written through contributions of all authors.

### Competing Interests

The authors declare no competing interests.

**Disclaimer**

The US Environmental Protection Agency funded this instrument development, but no formal review of these results has been conducted. This work has not been reviewed by US EPA. The views and conclusions expressed are strictly based on the evaluation of Aerodyne Research, Inc. scientists only.

### Acknowledgements

Development of the TILDAS-FD-EtO instrument with 413 m cell was funded by the US Environmental Protection Agency under SBIR award number 68HERC21C0047. The authors thank Jean-Marie Flaud and colleagues (Lafferty et al., 2013) for insights into high resolution Fourier transform infrared spectra for EtO.



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





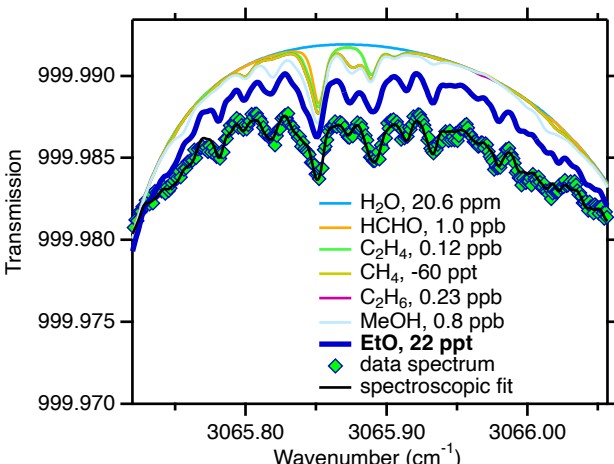

**Figure 1. Spectrum of EtO and other gaseous absorbers in the spectral window that are included in the spectroscopic fit. A measured spectrum (green diamonds, 24 hr average ambient spectrum, humidity-matched zeroes) is shown overlaid with the final fit (black trace). Individual fit components include water (H₂O), formaldehyde (HCHO), ethylene (C₂H₄), methane (CH₄), ethane (C₂H₆) and methanol (MeOH).**





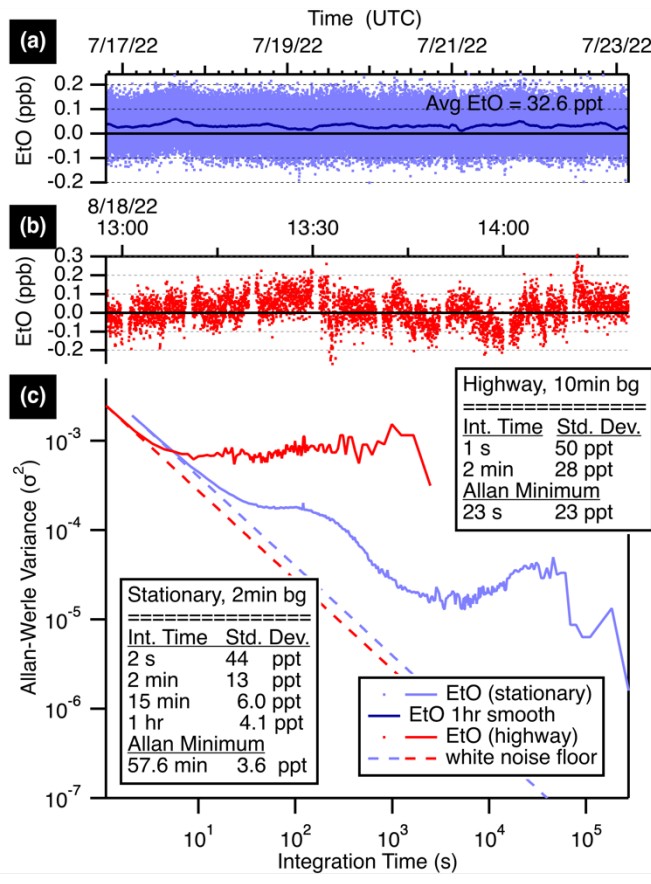

**Figure 2. Time series (a, b) and Allan-Werle variance plots (c) showing EtO precisions at various averaging times while stationary with 2-min autobackgrounds (blue), and while mobile on the highway with 10-min backgrounds (red).**

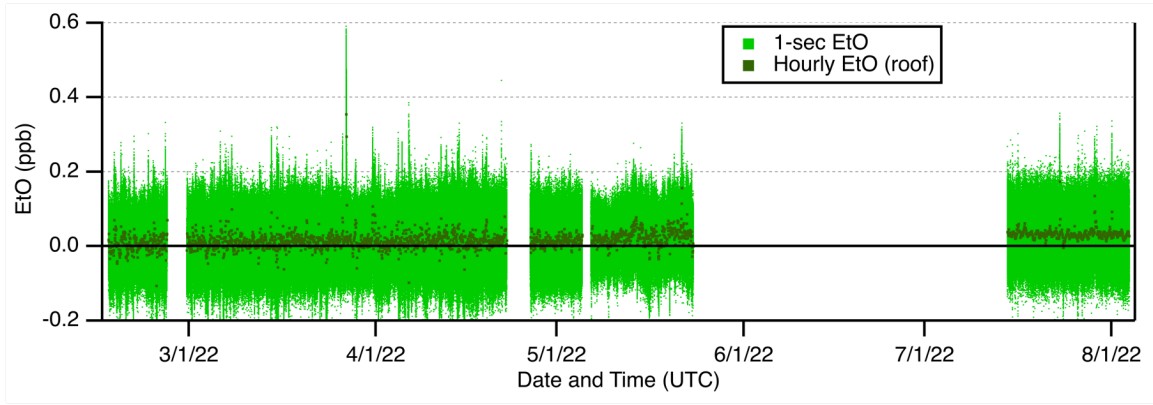

**Figure 3. Ambient ethylene oxide at a site in Billerica, Massachusetts, USA. Data at 1-second (pale green) is shown alongside hourly averages (dark green squares). Data prior to 6/2022 was acquired from a roof-top inlet with humidity-matched autobackgrounds every 5 minutes; data after 7/2022 was acquired from a 3-meter inlet with humidity-matched autobackgrounds every 2 minutes. Gaps in the time series are due to laboratory or field experiments.**




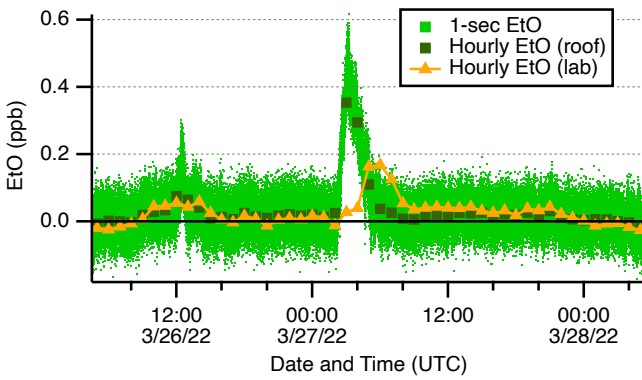


**Figure 4. Ethylene oxide events measured on the roof-top inlet. Outdoor data at 1-second data (pale green) is shown alongside hourly averages (dark green squares) averages. Laboratory air sampled prior to autobackgrounds (orange triangles) is shown.**

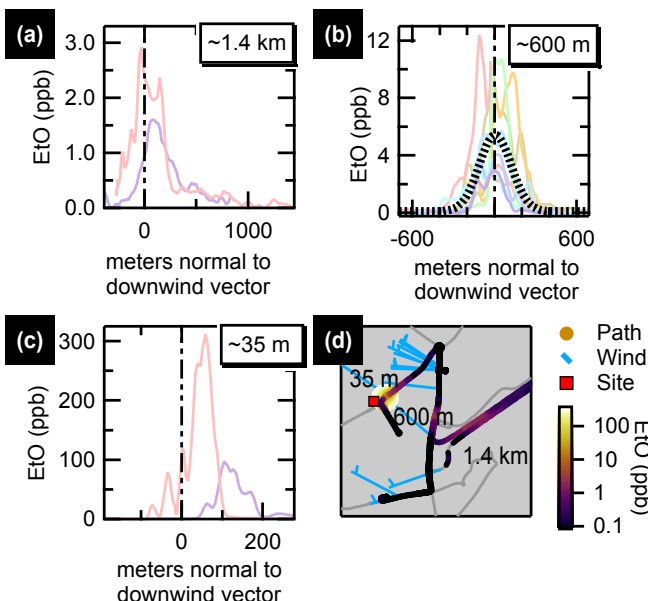

**Figure 5. Summary of transects downwind of Facility A. Transects are plotted normal to the wind vector for paths driven along 3**
**roads approximately 35 m (a), 600 m (b) and 1.4 km (c) downwind of Facility A. The average of 600 m transects (black dotted line) is shown for Panel B. A map (d) shows the facility location (red square) with the three main transect roads labelled by distance downwind. The driven path is colored and sized by EtO concentration. Wind vectors (blue) point into the wind.**