# Peer review of "Ethylene Oxide Monitor with Part-per-Trillion Precision for In-Situ Measurements"

_Atmospheric Measurement Techniques, 2022_

## Referee Comment (RC1)

**Review of Atmospheric Measurement Techniques Discussions Paper**
"Ethylene Oxide Monitor with Part-per-Trillion Precision for In-Situ Measurements"

by Tara Yacovitch, Christoph Dyroff, Joseph Roscioli, Conner Daube, J. Barry McManus, and Scott Herndon

This paper describes an instrument and associated measurement and data processing protocols for measurements of the important carcinogen trace gas ethylene oxide. As discussed, these measurements are extremely challenging due to the very low ambient concentrations at pptv levels and a number of potential spectroscopic interferences from higher concentration ambient constituents. This paper is highly relevant, the discussion is clearly written, and the measurements are carefully carried out. The ambient ethylene oxide enhancements downwind of two facilities known for operations involving this gas provide clear convincing evidence for the merits of this instrument as well as the instrument performance in real-world ambient conditions. This reviewer recommends final publication of this paper after the authors consider the following minor comments/suggestions to improve the paper quality further.

1. Would be informative to briefly describe the definition for the inhalation unit risk discussed in the Introduction.

2. Would also be informative to briefly describe in the Introduction that ethylene oxide ($C_2H_4O$, MW = 44.052 g/mole) is a cyclic three-membered ring structure with the O atom connected to both carbons.

3. The discussion of the simulations in Fig. 1 should be modified to indicate that you employed the Harrison et al. line parameters also for ethane here. The conditions of temperature and pressure should be included in figure caption. I am a little confused by the choice for the simulated concentrations included in Fig. 1. Shouldn't ambient levels of $CH_4$ around 2 ppm and $H_2O$ levels of 1 to 4% be used in these simulations or do these simulations represent the residual concentrations after subtracting the humidified matched background spectra? This should be discussed here.

4. The meaning of normalized in Fig. S1 should be included in the figure caption just as you did in the Supplement text on line 46. Also the spelling of "Mcmanus" on line 27 in the Supplement should be corrected to "McManus". Also maybe indicate why you get a normalized value up to 1.04 in Fig. S1. Is this due to noise or small inaccuracies in your polynomial baseline fitting here?

5. Indicate in the figure caption of Fig. S1 if the blue fit spectrum includes all the gases in the inset of Fig. 1?

6. The certified concentrations for EtO (1.075 ppm) and ethane (1.092 ppm) on line 76 needs to be reversed in accordance with the Analytical Results of Fig. S3. Also the X-axis labels in Fig. S2 in both cases needs to be corrected to 1.092 ppm in accordance with the

Analytical Results. Also, please explain in Fig. S2 why you label the left hand plot "Dry Calibration" as the $H_2O$ values here are actually larger than the right hand plot.

7.  On line 60 in the Supplement, you should consider either adding what is in the scrubber cartridge that removes EtO and not $H_2O$ or indicate this is proprietary.

8.  On line 87 in the discussion of dividing the subsequent sample spectra, it would be important to indicate if you employ the averaged background spectra over the ambient interval or do you use the updated background spectra for the subsequent ambient spectra? How much do these subsequent background spectra change (i.e., the difference of background spectra).

9.  Line 90, what scrubber breakthough are you referring to, breakthrough in EtO or $H_2O$? The text implies EtO breakthrough, but this should be spelled out.

10. Line 95 where you indicate the autobackground cycles, I am confused by the cycle values. Shouldn't the mobile measurements employ more frequent background measurements to capture the greater potential due to spatial changes in $H_2O$ and the reverse for stationary samples? Please further explain.

11. In Table S1 please indicate what * refers to in the Table next to the value 0.999

12. In Fig. 2c, you should add to the Y-axis label the units $ppb^2$

13. Line 103: I would change the wording " Measurements average down well" to something like" The variance improves with averaging time….", which better describes the plot Fig. 2c.

14. Line 107: You should reword " Optical alignment minimizes …" to something like " Adjustments to optical alignment …." Could small changes in the multipass highly dense spot pattern or resulting changes in optical cell noise also be a partially responsible?

15. In Fig. 2c, you should more clearly highlight in the plot the results for the EtO 1 hour smooth. As plotted, I have a hard time recognizing this 1 hour smooth. Are you referring to the portion of the variance between $10^3$ to $10^5$ sec? If so, you should darken this more in the plot.

16. On Line 108: I would think about rewording the statement " Continuous vibrations are less impactful..", as the red in-motion variance clearly shows reduced performance relative to the blue stationary performance. I think you are referring to the very large negative 0.5 ppb instantaneous deviations and not the more sustained red variance. Maybe adding a caveat to your statement?

17. It would be useful to provide an additional Fig. 3b plot showing only the hourly measurements with an expanded scale from say -0.05 to +0.05 ppb. This would highlight

better the two regimes. I just now saw this information is contained in your Fig. S4 and would leave it up to the co-authors to include a new Fig. 3b.

18. Line 128: It would be important to point out the importance of your observations that indoor laboratory air echoes outside air offset by 3 hours to highlight that a typical building ventilation system only minimally removes EtO by a factor of 2.

19. The back trajectory in Fig. S6 provides very useful information but the Google Street View inset really doesn't add anything. I would recommend providing a more convincing view of this facility (if you can legally show a picture of this sterilization facility) or remove the inset.

20.  In the Figures S4 showing facility A and the wind barbs, the facility A site indicator should be made larger in each case. Also the conventional definition of a wind barb indicates the direction from which the wind is blowing. The explanation in the caption of Fig. S7 and Fig. 5 indicating the wind barbs pointing into the wind is a little confusing given the conventional definition. This needs to be clarified.

---

## Community Comment (CC1)

**Response to Reviewer Comments for**

**"Ethylene Oxide Monitor with Part-per-Trillion Precision for In-Situ Measurements"**

**by Yacovitch et al.**

Reviewer comments are shown in blue, and responses are shown in black. **Bolded** text indicates text that has changed.

**Reviewer comments "RC1"**

https://doi.org/10.5194/amt-2022-294-RC1

This paper describes an instrument and associated measurement and data processing protocols for measurements of the important carcinogen trace gas ethylene oxide. As discussed, these measurements are extremely challenging due to the very low ambient concentrations at pptv levels and a number of potential spectroscopic interferences from higher concentration ambient constituents. This paper is highly relevant, the discussion is clearly written, and the measurements are carefully carried out. The ambient ethylene oxide enhancements downwind of two facilities known for operations involving this gas provide clear convincing evidence for the merits of this instrument as well as the instrument performance in real-world ambient conditions. This reviewer recommends final publication of this paper after the authors consider the following minor comments/suggestions to improve the paper quality further.

We thank the reviewer for their in-depth read and useful comments. We address them all below:

1. Would be informative to briefly describe the definition for the inhalation unit risk discussed in the Introduction.

We elaborate as follows:

The United States Environmental Protection Agency (EPA), through its Integrated Risk Information System (IRIS), has set an inhalation unit risk **(IUR)** for EtO at $3.0 \times 10^{-3}$ per $\mu g/m^3$ ($5.5 \times 10^{-3}$ per ppb), for adult increased cancer risk based on human data (US EPA, 2016). **The IUR is an upper-bound estimate of excess cancer risk from continuous exposure to a compound at 1 $\mu g/m^3$ in air (US EPA, 2022b). An IUR for EtO of $3.0 \times 10^{-3}$ per $\mu g/m^3$ implies that 3 excess cancer cases are expected to develop in 1000 people if exposed to 1 $\mu g/m^3$ (0.55 ppb) of EtO over a lifetime. Other risk estimates for different populations are included in the source EPA material (US EPA, 2016).**

2. Would also be informative to briefly describe in the Introduction that ethylene oxide ($C_2H_4O$, MW = 44.052 g/mole) is a cyclic three-membered ring structure with the O atom connected to both carbons.

"Ethylene oxide (EtO, also known as EO or oxirane) is a reactive **compound with a strained 3-member ether ring ($C_2H_4O$, CAS# 75-21-8, MW=44.05 g/mol).**"

> 3. The discussion of the simulations in Fig. 1 should be modified to indicate that you employed the Harrison et al. line parameters also for ethane here. The conditions of temperature and pressure should be included in figure caption. I am a little confused by the choice for the simulated concentrations included in Fig. 1. Shouldn't ambient levels of $CH_4$ around 2 ppm and $H_2O$ levels of 1 to 4% be used in these simulations or do these simulations represent the residual concentrations after subtracting the humidified matched background spectra? This should be discussed here.

We add the ethane line source, and move this statement directly after the HITRAN mention:

"[…] all molecules except EtO, **ethane** and methanol are from the HITRAN database (Gordon et al., 2017). **Ethane and methanol lines are based on experiments by Harrison et al. (2012).**"

Yes, the spectrum is a "zeroed" ambient spectrum, so ambient levels of $CH_4$, ethane and $H_2O$ are divided out of the fit. We explain in a few places:

Upon first mention of Figure 1:

"[…] Figure 1. **This figure fits an ambient spectrum divided by a scrubber-zeroed spectrum, such that all species except for EtO are near-zero (see Section 2.3).**"

In the Figure 1 caption:

" Figure 1. Spectrum of EtO and other gaseous absorbers in the spectral window that are included in the spectroscopic fit. A measured spectrum (green diamonds, 24 hr average ambient spectrum, humidity-matched **scrubber** zeroes) is shown overlaid with the final fit (black trace). Individual fit components include water ($H_2O$), formaldehyde (HCHO), ethylene ($C_2H_4$), methane ($CH_4$), ethane ($C_2H_6$) and methanol (MeOH). **This figure fits an ambient spectrum divided by a scrubber-zeroed spectrum, such that all species except for EtO are near-zero (see Section 2.3).**"

In Section 2.3:

The use of scrubbed air provides a near-humidity match between sample and background spectra, effectively flattening out the curvature of the baseline present under the EtO lines due to strong neighboring water absorptions. **We have not extensively tested whether the scrubber decreases the other species measured in the fit (HCHO, $C_2H_6$, $C_2H_4$, $CH_4$, etc.), but they appear in the divided ambient spectra with near-zero concentrations (Figure 1). For species with significant ambient backgrounds like $CH_4$, this is indicates that the scrubber is non-destructive to $CH_4$.**

> 4. The meaning of normalized in Fig. S1 should be included in the figure caption just as you did in the Supplement text on line 46. Also the spelling of "Mcmanus" on line 27 in the Supplement should be corrected to "McManus". Also maybe indicate why you get a normalized value up to 1.04 in Fig. S1. Is this due to noise or small inaccuracies in your polynomial baseline fitting here?

The capitalization of "McManus" has been corrected.

We have added a short discussion of this figure and moved it in the SI under the "S4.1 Facility A" header, since it is not meant to illustrate the 0-1 transmission normalization procedure described in the SI section "S1.1 Optical setup […]" section

**The archived spectra can be used to unambiguously spectrally fingerprint the EtO observed at these facilities. For example, Figure S10 (top) shows raw measured signal out of plume (black) overlaid on signal in-plume (gold area) at Facility A. In Figure S10 (bottom), we manually divide the in-plume and out-of-plume spectra to reveal the spectral signature of EtO. Line scars at the positions of the water lines are observable (green spikes) due to slight variations in laser peak position. The blue line is a transmission simulation of EtO only, and clearly matches the experimental result.**

[Figure]

**Figure S10. Summary spectra comparing instantaneous Facility A measurement of 747 ppb (in-plume, gold) to an out-of plume spectrum. The top shows signal as a function of wavenumber, with EtO contributions highlighted in yellow. The bottom shows a divided spectra in-plume/out-of-plume (green) and transmission simulation of EtO only.**

5. Indicate in the figure caption of Fig. S1 if the blue fit spectrum includes all the gases in the inset of Fig. 1?

It does not. See discussion above

6. The certified concentrations for EtO (1.075 ppm) and ethane (1.092 ppm) on line 76 needs to be reversed in accordance with the Analytical Results of Fig. S3. Also the X-

We have corrected this concentration typo in the text. We have confirmed that the correct tank concentrations were used in the calibration workups, including those shown in Fig S2. The axis Labels for Fig S2 have been corrected to read 1.092.

The water concentrations were in scientific notation, and hard to parse (6E4 ppb for the dry calibration; 1.5E7 ppb for the standard addition). We have changed both to percent water (0.006% for the dry calibration vs 1.5% for the standard addition) for clarity.

7. On line 60 in the Supplement, you should consider either adding what is in the scrubber cartridge that removes EtO and not $H_2O$ or indicate this is proprietary.

We add the following to the SI:

For humidity-matched zeroes, a parallel flow path is set up, with a length-matched piece of tubing and **6" by 1"** scrubber cartridge isolated by two solenoid valves. **We use a manganese dioxide/copper oxide catalyst as scrubber: Carulite 500® (Carus LLC), heated to 150 C.**

8. On line 87 in the discussion of dividing the subsequent sample spectra, it would be important to indicate if you employ the averaged background spectra over the ambient interval or do you use the updated background spectra for the subsequent ambient spectra? How much do these subsequent background spectra change (i.e., the difference of background spectra).

We alter the text to specify that we use the nearest prior background spectrum:

"**Each acquired** background spectrum is used to **divide sample spectra for the subsequent period,** […]"

We have analyzed a 24 hr period of data on 7/20/2022 with 2-minute backgrounds. The average background-to-background delta is 21 ppt. For reference, the average ambient EtO on this day was 29 ppt (sdev = 48 ppt).

Each background is a 10-second average. The collected raw background spectra, with their deep water lines, were each divided by the average 24hr background spectrum prior to refitting, so that the input spectra had similar characteristics (flattened baseline about the EtO lines) as the fit sample spectra. A spectral fit of these data then yields a measured "zero" EtO for each background.

The average background-to-background delta of 21 ppt is on the same scale as the expected instrument performance on the timescale of these backgrounds: the Figure 2 Allan-Werle variance plot shows a 10 sec 1σ precision of 20.8 ppt; and a 2 min 1σ precision of 13 ppt. This implies that for 7/20/2022, background-to-background drift is being optimally mitigated at a 2-min zero cycle, which can also be seen from the variance plot itself.

9. Line 90, what scrubber breakthough are you referring to, breakthrough in EtO or $H_2O$? The text implies EtO breakthrough, but this should be spelled out.

We clarify:

"Laboratory experiments suggest scrubber **EtO** breakthrough on the scale of 3% is possible (3-5 SLPM flow rates) at high mixing ratios (hundreds of ppb). Indeed, mobile near-source measurements have shown such **EtO** breakthrough […]"

10. Line 95 where you indicate the autobackground cycles, I am confused by the cycle values. Shouldn't the mobile measurements employ more frequent background measurements to capture the greater potential due to spatial changes in $H_2O$ and the reverse for stationary samples? Please further explain.

The scrubber provides good but imperfect humidity matching, and so we continuously fit water in between zeroes. We have thus not found spatial changes in $H_2O$ on the timescale of a mobile zero to be of concern. We rework this paragraph to further explain the basis for our zero timing:

"The frequency of autobackgrounds is chosen to match the sampling strategy. **Mobile** measurements aimed at capturing plumes (enhancements over background **lasting typically 1-3 minutes**) use a 5- to 15-minute autobackground cycle. **This is a practical decision that reduces the chance of a zero interfering with a plume during a downwind transect of a facility, and is defensible as we typically are less concerned with time averaging and ppt-level baseline drift during near-source measurements.** Stationary sampling of background concentrations, on the other hand, **yields best long-term averaging with a 2-minute cycle.**"

11. In Table S1 please indicate what * refers to in the Table next to the value 0.999

The original note indicated that the first dry calibration of the measurements was an outlier at 0.895 (low). We remove this asterisk, as the remaining calibrations are still within a week-long period.

12. In Fig. 2c, you should add to the Y-axis label the units $ppb^2$

The Y axis has been relabeled:

"**Allan-Werle Variance: EtO $\sigma^2$ ($ppb^2$)**"

13. Line 103: I would change the wording " Measurements average down well" to something like" The variance improves with averaging time....", which better describes the plot Fig. 2c.

We reword:

"**The precision improves with averaging time,** […]"

14. Line 107: You should reword " Optical alignment minimizes ..." to something like " Adjustments to optical alignment ...." Could small changes in the multipass highly dense spot pattern or resulting changes in optical cell noise also be a partially responsible?

The optical cell's mirrors are fixed in position and orientation, and so the spot pattern itself is very robust. An early exit of the laser beam from the cell is possible with very poor alignment of the input mirrors, but this is a dramatic effect, and not something that we have observed in motion. The main culprit is the focusing objective, as we describe in the text.

We further reword to:

"The instrument shows sensitivity to truck motion, particularly quick turns or stops which manifest as negative deviations in mixing ratio on the order of 0.5 ppb. **Optimizing optical alignment minimizes but does not eliminate these effects, which are largely attributed to strain on the laser focusing objective.**"

> 15. In Fig. 2c, you should more clearly highlight in the plot the results for the EtO 1 hour smooth. As plotted, I have a hard time recognizing this 1 hour smooth. Are you referring to the portion of the variance between $10^3$ to $10^5$ sec? If so, you should darken this more in the plot.

The 1-hr smooth is for the stationary data due to the density of data shown. We have changed the line type and figure caption to clarify.

[Figure]

**Figure 2.** Time series (a, b) and Allan-Werle variance plots (c) showing EtO precisions at various averaging times while stationary with 2-min autobackgrounds (blue), and while mobile on the highway with 10-min backgrounds (red). **The stationary data (a) averages to 32.6 ppt EtO, with a 1-hour smooth (dotted line) shown.**

> 16. On Line 108: I would think about rewording the statement "Continuous vibrations are less impactful..", as the red in-motion variance clearly shows reduced performance relative to the blue stationary performance. I think you are referring to the very large negative 0.5 ppb instantaneous deviations and not the more sustained red variance. Maybe adding a caveat to your statement?

We rephrase: "Continuous vibrations **do not manifest as negative deviations, instead impacting the overall noise.**"

17. It would be useful to provide an additional Fig. 3b plot showing only the hourly measurements with an expanded scale from say -0.05 to +0.05 ppb. This would highlight better the two regimes. I just now saw this information is contained in your Fig. S4 and would leave it up to the co-authors to include a new Fig. 3b.

We reference Figure S3 (old Figure S4) in the text explicitly and add a panel to panel to Figure 3
showing a monthly box plot:

[Figure]

**Figure 3. Ambient ethylene oxide at a site in Billerica, Massachusetts, USA. Panel (a): Data at 1-second (pale green) is shown alongside hourly averages (dark green squares). Panel (b): Monthly box plot showing the median, 25th and 75th quartiles, with whiskers extending to the 5th and 95th percentiles. Data prior to 6/2022 were acquired from a roof-top inlet with humidity-matched autobackgrounds every 30 minutes; data after 7/2022 were acquired from a 3-meter inlet with humidity-matched autobackgrounds every 2 minutes. Gaps in the time series are due to laboratory or field experiments.**

18. Line 128: It would be important to point out the importance of your observations that indoor laboratory air echoes outside air offset by 3 hours to highlight that a typical building ventilation system only minimally removes EtO by a factor of 2.

"The laboratory air shows an "echo" of the outdoor EtO event ~3 hours delayed**, and slightly
broadened, with a maximum concentration of 168 ppt**, which we attribute to the building's
ventilation system gradually mixing in outdoor air. **This observation highlights the fact that
indoor air quality is directly impacted by outdoor EtO concentrations.**"

19. The back trajectory in Fig. S6 provides very useful information but the Google Street View inset really doesn't add anything. I would recommend providing a more convincing view of this facility (if you can legally show a picture of this sterilization facility) or remove the inset.

We choose not to publish facility images, but have added a panel next to Figure S6 that shows the
location of both potential EtO facilities referenced in US EPA, 2022b, along with the location of
the rooftop measurements.

[Figure]

Figure S5. Left: Detail of back trajectory (NOAA Air Resources Laboratory). The orange arrow indicates location of one potential EtO source in the state. A roadside-view of this facility's signage is inset (© Google Street View), with "EO deliveries" noted. **Right: Location of rooftop measurements (black marker) and two potential EtO facilities (red markers) in the state of Massachusetts. Map outlines from NOAA (NOAA, 2013).**

20. In the Figures S4 showing facility A and the wind barbs, the facility A site indicator should be made larger in each case. Also the conventional definition of a wind barb indicates the direction from which the wind is blowing. The explanation in the caption of Fig. S7 and Fig. 5 indicating the wind barbs pointing into the wind is a little confusing given the conventional definition. This needs to be clarified.

Facility transect figures have had the facility markers enlarged, as in the example below.

[Figure]

We follow the convention for wind barbs (e.g. https://www.weather.gov/hfo/windbarbinfo). We rephrase in the captions for SI figures and Figure 5: "[…] **with wind barbs tethered to the truck path, and feather end of the staff pointing into the wind.**"

---

## Author Response (AR1)

**Response to Reviewer Comments for**

**"Ethylene Oxide Monitor with Part-per-Trillion Precision for In-Situ Measurements"**

**by Yacovitch et al.**

Reviewer comments are shown in blue, and responses are shown in black. **Bolded** text indicates text that has changed.

**Reviewer comments "RC1"**

https://doi.org/10.5194/amt-2022-294-RC1

This paper describes an instrument and associated measurement and data processing protocols for measurements of the important carcinogen trace gas ethylene oxide. As discussed, these measurements are extremely challenging due to the very low ambient concentrations at pptv levels and a number of potential spectroscopic interferences from higher concentration ambient constituents. This paper is highly relevant, the discussion is clearly written, and the measurements are carefully carried out. The ambient ethylene oxide enhancements downwind of two facilities known for operations involving this gas provide clear convincing evidence for the merits of this instrument as well as the instrument performance in real-world ambient conditions. This reviewer recommends final publication of this paper after the authors consider the following minor comments/suggestions to improve the paper quality further.

We thank the reviewer for their in-depth read and useful comments. We address them all below:

1. Would be informative to briefly describe the definition for the inhalation unit risk discussed in the Introduction.

We elaborate as follows:

The United States Environmental Protection Agency (EPA), through its Integrated Risk Information System (IRIS), has set an inhalation unit risk **(IUR)** for EtO at $3.0 \times 10^{-3}$ per $\mu g/m^3$ ($5.5 \times 10^{-3}$ per ppb), for adult increased cancer risk based on human data (US EPA, 2016). **The IUR is an upper-bound estimate of excess cancer risk from continuous exposure to a compound at 1 $\mu g/m^3$ in air (US EPA, 2022b). An IUR for EtO of $3.0 \times 10^{-3}$ per $\mu g/m^3$ implies that 3 excess cancer cases are expected to develop in 1000 people if exposed to 1 $\mu g/m^3$ (0.55 ppb) of EtO over a lifetime. Other risk estimates for different populations are included in the source EPA material (US EPA, 2016).**

2. Would also be informative to briefly describe in the Introduction that ethylene oxide ($C_2H_4O$, MW = 44.052 g/mole) is a cyclic three-membered ring structure with the O atom connected to both carbons.

"Ethylene oxide (EtO, also known as EO or oxirane) is a reactive **compound with a strained 3-member ether ring ($C_2H_4O$, CAS# 75-21-8, MW=44.05 g/mol).**"

> 3. The discussion of the simulations in Fig. 1 should be modified to indicate that you employed the Harrison et al. line parameters also for ethane here. The conditions of temperature and pressure should be included in figure caption. I am a little confused by the choice for the simulated concentrations included in Fig. 1. Shouldn't ambient levels of $CH_4$ around 2 ppm and $H_2O$ levels of 1 to 4% be used in these simulations or do these simulations represent the residual concentrations after subtracting the humidified matched background spectra? This should be discussed here.

We add the ethane line source, and move this statement directly after the HITRAN mention:

"[…] all molecules except EtO, **ethane** and methanol are from the HITRAN database (Gordon et al., 2017). **Ethane and methanol lines are based on experiments by Harrison et al. (2012).**"

Yes, the spectrum is a "zeroed" ambient spectrum, so ambient levels of $CH_4$, ethane and $H_2O$ are divided out of the fit. We explain in a few places:

Upon first mention of Figure 1:

"[…] Figure 1. **This figure fits an ambient spectrum divided by a scrubber-zeroed spectrum, such that all species except for EtO are near-zero (see Section 2.3).**"

In the Figure 1 caption:

" Figure 1. Spectrum of EtO and other gaseous absorbers in the spectral window that are included in the spectroscopic fit. A measured spectrum (green diamonds, 24 hr average ambient spectrum, humidity-matched **scrubber** zeroes) is shown overlaid with the final fit (black trace). Individual fit components include water ($H_2O$), formaldehyde (HCHO), ethylene ($C_2H_4$), methane ($CH_4$), ethane ($C_2H_6$) and methanol (MeOH). **This figure fits an ambient spectrum divided by a scrubber-zeroed spectrum, such that all species except for EtO are near-zero (see Section 2.3).**"

In Section 2.3:

The use of scrubbed air provides a near-humidity match between sample and background spectra, effectively flattening out the curvature of the baseline present under the EtO lines due to strong neighboring water absorptions. **We have not extensively tested whether the scrubber decreases the other species measured in the fit (HCHO, $C_2H_6$, $C_2H_4$, $CH_4$, etc.), but they appear in the divided ambient spectra with near-zero concentrations (Figure 1). For species with significant ambient backgrounds like $CH_4$, this is indicates that the scrubber is non-destructive to $CH_4$.**

> 4. The meaning of normalized in Fig. S1 should be included in the figure caption just as you did in the Supplement text on line 46. Also the spelling of "Mcmanus" on line 27 in the Supplement should be corrected to "McManus". Also maybe indicate why you get a normalized value up to 1.04 in Fig. S1. Is this due to noise or small inaccuracies in your polynomial baseline fitting here?

The capitalization of "McManus" has been corrected.

We have added a short discussion of this figure and moved it in the SI under the "S4.1 Facility A" header, since it is not meant to illustrate the 0-1 transmission normalization procedure described in the SI section "S1.1 Optical setup […]" section

**The archived spectra can be used to unambiguously spectrally fingerprint the EtO observed at these facilities. For example, Figure S10 (top) shows raw measured signal out of plume (black) overlaid on signal in-plume (gold area) at Facility A. In Figure S10 (bottom), we manually divide the in-plume and out-of-plume spectra to reveal the spectral signature of EtO. Line scars at the positions of the water lines are observable (green spikes) due to slight variations in laser peak position. The blue line is a transmission simulation of EtO only, and clearly matches the experimental result.**

[Figure]

**Figure S10. Summary spectra comparing instantaneous Facility A measurement of 747 ppb (in-plume, gold) to an out-of plume spectrum. The top shows signal as a function of wavenumber, with EtO contributions highlighted in yellow. The bottom shows a divided spectra in-plume/out-of-plume (green) and transmission simulation of EtO only.**

5. Indicate in the figure caption of Fig. S1 if the blue fit spectrum includes all the gases in the inset of Fig. 1?

It does not. See discussion above

6. The certified concentrations for EtO (1.075 ppm) and ethane (1.092 ppm) on line 76 needs to be reversed in accordance with the Analytical Results of Fig. S3. Also the X-

We have corrected this concentration typo in the text. We have confirmed that the correct tank concentrations were used in the calibration workups, including those shown in Fig S2. The axis Labels for Fig S2 have been corrected to read 1.092.

The water concentrations were in scientific notation, and hard to parse (6E4 ppb for the dry calibration; 1.5E7 ppb for the standard addition). We have changed both to percent water (0.006% for the dry calibration vs 1.5% for the standard addition) for clarity.

We add the following to the SI:

For humidity-matched zeroes, a parallel flow path is set up, with a length-matched piece of tubing and **6" by 1"** scrubber cartridge isolated by two solenoid valves. **We use a manganese dioxide/copper oxide catalyst as scrubber: Carulite 500® (Carus LLC), heated to 150 C.**

We alter the text to specify that we use the nearest prior background spectrum:

"**Each acquired** background spectrum is used to **divide sample spectra for the subsequent period,** […]"

We have analyzed a 24 hr period of data on 7/20/2022 with 2-minute backgrounds. The average background-to-background delta is 21 ppt. For reference, the average ambient EtO on this day was 29 ppt (sdev = 48 ppt).

Each background is a 10-second average. The collected raw background spectra, with their deep water lines, were each divided by the average 24hr background spectrum prior to refitting, so that the input spectra had similar characteristics (flattened baseline about the EtO lines) as the fit sample spectra. A spectral fit of these data then yields a measured "zero" EtO for each background.

The average background-to-background delta of 21 ppt is on the same scale as the expected instrument performance on the timescale of these backgrounds: the Figure 2 Allan-Werle variance plot shows a 10 sec 1σ precision of 20.8 ppt; and a 2 min 1σ precision of 13 ppt. This implies that for 7/20/2022, background-to-background drift is being optimally mitigated at a 2-min zero cycle, which can also be seen from the variance plot itself.

We clarify:

"Laboratory experiments suggest scrubber **EtO** breakthrough on the scale of 3% is possible (3-5 SLPM flow rates) at high mixing ratios (hundreds of ppb). Indeed, mobile near-source measurements have shown such **EtO** breakthrough […]"

10. Line 95 where you indicate the autobackground cycles, I am confused by the cycle values. Shouldn't the mobile measurements employ more frequent background measurements to capture the greater potential due to spatial changes in $H_2O$ and the reverse for stationary samples? Please further explain.

The scrubber provides good but imperfect humidity matching, and so we continuously fit water in between zeroes. We have thus not found spatial changes in $H_2O$ on the timescale of a mobile zero to be of concern. We rework this paragraph to further explain the basis for our zero timing:

"The frequency of autobackgrounds is chosen to match the sampling strategy. **Mobile** measurements aimed at capturing plumes (enhancements over background **lasting typically 1-3 minutes**) use a 5- to 15-minute autobackground cycle. **This is a practical decision that reduces the chance of a zero interfering with a plume during a downwind transect of a facility, and is defensible as we typically are less concerned with time averaging and ppt-level baseline drift during near-source measurements.** Stationary sampling of background concentrations, on the other hand, **yields best long-term averaging with a 2-minute cycle.**"

11. In Table S1 please indicate what * refers to in the Table next to the value 0.999

The original note indicated that the first dry calibration of the measurements was an outlier at 0.895 (low). We remove this asterisk, as the remaining calibrations are still within a week-long period.

12. In Fig. 2c, you should add to the Y-axis label the units $ppb^2$

The Y axis has been relabeled:

"**Allan-Werle Variance: EtO $\sigma^2$ ($ppb^2$)**"

13. Line 103: I would change the wording " Measurements average down well" to something like" The variance improves with averaging time....", which better describes the plot Fig. 2c.

We reword:

"**The precision improves with averaging time,** […]"

14. Line 107: You should reword " Optical alignment minimizes ..." to something like " Adjustments to optical alignment ...." Could small changes in the multipass highly dense spot pattern or resulting changes in optical cell noise also be a partially responsible?

The optical cell's mirrors are fixed in position and orientation, and so the spot pattern itself is very robust. An early exit of the laser beam from the cell is possible with very poor alignment of the input mirrors, but this is a dramatic effect, and not something that we have observed in motion. The main culprit is the focusing objective, as we describe in the text.

We further reword to:

"The instrument shows sensitivity to truck motion, particularly quick turns or stops which manifest as negative deviations in mixing ratio on the order of 0.5 ppb. **Optimizing optical alignment minimizes but does not eliminate these effects, which are largely attributed to strain on the laser focusing objective.**"

15. In Fig. 2c, you should more clearly highlight in the plot the results for the EtO 1 hour smooth. As plotted, I have a hard time recognizing this 1 hour smooth. Are you referring to the portion of the variance between $10^3$ to $10^5$ sec? If so, you should darken this more in the plot.

The 1-hr smooth is for the stationary data due to the density of data shown. We have changed the line type and figure caption to clarify.

[Figure]

**Figure 2.** Time series (a, b) and Allan-Werle variance plots (c) showing EtO precisions at various averaging times while stationary with 2-min autobackgrounds (blue), and while mobile on the highway with 10-min backgrounds (red). **The stationary data (a) averages to 32.6 ppt EtO, with a 1-hour smooth (dotted line) shown.**

16. On Line 108: I would think about rewording the statement "Continuous vibrations are less impactful..", as the red in-motion variance clearly shows reduced performance relative to the blue stationary performance. I think you are referring to the very large negative 0.5 ppb instantaneous deviations and not the more sustained red variance. Maybe adding a caveat to your statement?

We rephrase: "Continuous vibrations **do not manifest as negative deviations, instead impacting the overall noise.**"

17. It would be useful to provide an additional Fig. 3b plot showing only the hourly
measurements with an expanded scale from say -0.05 to +0.05 ppb. This would highlight
better the two regimes. I just now saw this information is contained in your Fig. S4 and
would leave it up to the co-authors to include a new Fig. 3b.

We reference Figure S3 (old Figure S4) in the text explicitly and add a panel to panel to Figure 3
showing a monthly box plot:

[Figure]

**Figure 3. Ambient ethylene oxide at a site in Billerica, Massachusetts, USA. Panel (a): Data at 1-
second (pale green) is shown alongside hourly averages (dark green squares). Panel (b): Monthly box
plot showing the median, 25th and 75th quartiles, with whiskers extending to the 5th and 95th
percentiles. Data prior to 6/2022 were acquired from a roof-top inlet with humidity-matched
autobackgrounds every 30 minutes; data after 7/2022 were acquired from a 3-meter inlet with
humidity-matched autobackgrounds every 2 minutes. Gaps in the time series are due to laboratory
or field experiments.**

18. Line 128: It would be important to point out the importance of your observations that
indoor laboratory air echoes outside air offset by 3 hours to highlight that a typical
building ventilation system only minimally removes EtO by a factor of 2.

"The laboratory air shows an "echo" of the outdoor EtO event ~3 hours delayed**, and slightly
broadened, with a maximum concentration of 168 ppt**, which we attribute to the building's
ventilation system gradually mixing in outdoor air. **This observation highlights the fact that
indoor air quality is directly impacted by outdoor EtO concentrations.**"

19. The back trajectory in Fig. S6 provides very useful information but the Google Street
View inset really doesn't add anything. I would recommend providing a more convincing
view of this facility (if you can legally show a picture of this sterilization facility) or
remove the inset.

We choose not to publish facility images, but have added a panel next to Figure S6 that shows the
location of both potential EtO facilities referenced in US EPA, 2022b, along with the location of
the rooftop measurements.

[Figure]

Figure S5. Left: Detail of back trajectory (NOAA Air Resources Laboratory). The orange arrow indicates location of one potential EtO source in the state. A roadside-view of this facility's signage is inset (© Google Street View), with "EO deliveries" noted. **Right: Location of rooftop measurements (black marker) and two potential EtO facilities (red markers) in the state of Massachusetts. Map outlines from NOAA (NOAA, 2013).**

20. In the Figures S4 showing facility A and the wind barbs, the facility A site indicator should be made larger in each case. Also the conventional definition of a wind barb indicates the direction from which the wind is blowing. The explanation in the caption of Fig. S7 and Fig. 5 indicating the wind barbs pointing into the wind is a little confusing given the conventional definition. This needs to be clarified.

Facility transect figures have had the facility markers enlarged, as in the example below.

[Figure]

We follow the convention for wind barbs (e.g. https://www.weather.gov/hfo/windbarbinfo). We rephrase in the captions for SI figures and Figure 5: "[…] **with wind barbs tethered to the truck path, and feather end of the staff pointing into the wind.**"

**Response to Reviewer Comments for**

**"Ethylene Oxide Monitor with Part-per-Trillion Precision for In-Situ Measurements"**

**by Yacovitch et al.**

Reviewer comments are shown in blue, and responses are shown in black.

**Reviewer comments "RC2"**

https://doi.org/10.5194/amt-2022-294-RC2

Review of Yakovitch et al., AMTD

This is a nice concise paper that describes the TILDAS-FD-EtO analyzer for ethylene oxide.

This is a really difficult measurement (due to low conc and potential interferences) and the authors give a good description here.

The large enhancements from two Facilities observed at the lab are very interesting and chasing down those sources with the mobile lab is particularly impressive.

I only have a couple of minor comments below.

We thank the reviewer for their in-depth read and useful comments. We address them all below:

Line 7. I don't think the cell is 413m long. I think you mean a cell with a 413 m path length.
Indeed. We have reworded the sentence.

Line 55. Whats the cell pressure? Add it here.
We added the sentence: The sample pressure was maintained between 20 Torr (26 mbar) and 30 Torr (40 mbar) throughout the experiments described in this paper.

line 103. "averages down well" is something I would say but it is a bit informal. Maybe reword.
Agreed. We have reworded the sentence.

Line 129. Hysplit is a model not an engine.

We have reworded to model.

Is the 22 ppt sd real signal or noise? Maybe include the averaging time?  Maybe clarify what you explicitly mean here.

Yes, the 18 ppt is the average of the hourly data over the entire period. This was explained near original line 115 where these values are first mentioned. The instrument noise as a function of averaging time was explained in section Instrument Performance.

We have clarified further by adding the following sentence in Section Ambient Measurements: The standard deviations given reflect the combination of instrument noise as described above and the variability of EtO in ambient air.

We have also modified Figure 3 and added a box plot to it. This box plot shows the monthly median and percentiles. Furthermore, Figure S3 was added to show histograms of winter/spring and summer hourly average EtO values.

I think Fig S1 is especially helpful to the discussion. Would you consider moving it (or something similar) to the main text?

We have decided to leave this figure in the supplemental information.